# Are there ethnic and religious variations in uptake of bowel cancer screening? A retrospective cohort study among 1.7 million people in Scotland

Christine Campbell ,[1] Anne Douglas,[1] Linda Williams,[1] Geneviève Cezard,[2] David H Brewster ,[1] Duncan Buchanan,[3] Kathryn Robb,[4] Greig Stanners,[5] David Weller,[1] Robert JC Steele,[6] Markus Steiner,[7] Raj Bhopal[1]

For numbered affiliations see end of article.

Correspondence to
Dr Christine Campbell;
Christine.Campbell@ed.ac.uk

## ABSTRACT

**Objective** Cancer screening should be equitably accessed by all populations. Uptake of colorectal cancer screening was examined using the Scottish Health and Ethnicity Linkage Study that links the Scottish Census 2001 to health data by individual-level self-reported ethnicity and religion.

**Setting** Data on 1.7 million individuals in two rounds of the Scottish Bowel Cancer Screening Programme (2007–2013) were linked to the 2001 Census using the Scottish Community Health Index number.

**Main outcome measure** Uptake of colorectal cancer screening, reported as age-adjusted risk ratios (RRs) by ethnic group and religion were calculated for men and women with 95% CI.

**Results** In the first, incidence screening round, compared with white Scottish men, Other White British (RR 109.6, 95% CI 108.8 to 110.3) and Chinese (107.2, 95% CI 102.8 to 111.8) men had higher uptake. In contrast, men of all South Asian groups had lower uptake (Indian RR 80.5, 95% CI 76.1 to 85.1; Pakistani RR 65.9, 95% CI 62.7 to 69.3; Bangladeshi RR 76.6, 95% CI 63.9 to 91.9; Other South Asian RR 88.6, 95% CI 81.8 to 96.1). Comparable patterns were seen among women in all ethnic groups, for example, Pakistani (RR 55.5, 95% CI 52.5 to 58.8). Variation in uptake was also observed by religion, with lower rates among Hindu (RR (95%CI): 78.4 (71.8 to 85.6)), Muslim (69.5 (66.7 to 72.3)) and Sikh (73.4 (67.1 to 80.3)) men compared with the reference population (Church of Scotland), with similar variation among women: lower rates were also seen among those who reported being Jewish, Roman Catholic or with no religion.

**Conclusions** There are important variations in uptake of bowel cancer screening by ethnic group and religion in Scotland, for both sexes, that require further research and targeted interventions.

## BACKGROUND

Bowel cancer is the third most commonly diagnosed cancer in both men and women in Scotland.[1] Bowel cancer screening using the faecal occult blood test (FOBt) was started across all National Health Service (NHS)

## Strengths and limitations of this study

► The most fine-grained analyses of bowel screening uptake by ethnicity reported to date, using a nationally tested classification of ethnic groups.

► The study benefits from high overall linkage rates of census and National Health Service Community Health Index numbers, with a large national population, and a high linkage rate with Bowel Screening data.

► The small numbers of outcomes for some non-white populations has required aggregation of data for some ethnic groups, restricting reporting of invasive cancer for some ethnic groups due to potentially disclosive numbers.

► Patterns of immigration to Scotland over the last 18 years have changed, in particular among those from Eastern Europe, and we do not report bowel screening uptake among these populations.

► The reported screening uptake rates are descriptive and not explanatory: although we adjusted for determinants of ethnic inequalities in bowel screening such as socio-economic status and UK birth, these made little difference to the patterns observed, and further potential mechanisms need to be explored.

boards (health authorities) in Scotland between June 2007 and December 2009, with those aged between 50 and 74 years and registered with a general practice invited to participate every 2 years.[2] Routine use of a faecal immunochemical test (FIT) was introduced in November 2017. Although progress has been made, substantial variation in uptake is still observed by deprivation:[3] however, variation by ethnicity in Scotland has not been studied.

There is growing recognition of the challenges to minimising inequalities in cancer outcomes in minority ethnic populations across the UK. Recent work has demonstrated lower awareness of the breast and cervical

screening programmes compared with White survey participants and very low (less than 30% of respondents) awareness of bowel screening overall.[4] Lower attendance among Asian invitees in the UK Flexible Sigmoidoscopy Trial has also been reported.[5] The reasons for these differences are likely to include the approach of services, cultural beliefs and attitudes, and health communication and literacy barriers.[6 7]

Reporting of inequalities in uptake of cancer screening by minority ethnic group has been limited by a failure in most health systems to routinely code ethnicity accurately. As a consequence, our understanding is based on area-based measures,[8 9] responders to surveys including items on ethnicity,[5] or name recognition software (eg, Nam Pehchan).[10] Existing evidence is further limited by the use of very broad categories for ethnicity, for example, Indian subcontinent[8]; white/black/Asian[5]; Hindu-Gujerati/Hindu-Other/Muslim/Sikh/Other Asian.[10] Within these constraints, variation in uptake has been observed internationally,[11] in the FOBt bowel screening pilots in England,[10 12 13] and has been reported in the English Screening Programme.[14] UK bowel screening databases (including Scotland) do not routinely include an ethnic code[15] so reported estimated uptake rates by ethnicity are based on area-level characteristics rather than individual-level data. Findings from other parts of the UK may not be generalisable due to differences in composition of ethnic minority groups and religious affiliations, cultural background and service provision. A better understanding of both screening uptake and screening outcomes, analysed by minority ethnic groups and by religion, has the potential to inform more targeted education and informed choice strategies (although recognising that ethnicity, religion and cultural background are overlapping although not synonymous identities).

This study made use of a unique UK resource, the Scottish Health and Ethnicity Linkage Study (SHELS): linkage of the 2001 Census in Scotland (with individual level self-reported ethnicity, country of birth, religion and a range of sociodemographic characteristics), with the Community Health Index (CHI) register number and through that to other health databases.[16 17] Linkage to the breast screening programme dataset enabled SHELS[18] to demonstrate lower uptake of breast screening among minority ethnic women in Scotland, even when adjusted for several confounding factors. The primary aim of this paper was to describe bowel cancer screening uptake rates in detail by self-reported ethnic group, including White Scottish, other white British, white Irish, other White, Indian, Pakistani, Bangladeshi, other South Asian, Caribbean, African Other Black, Chinese, in addition to self-reported religious affiliation. Further, as previous SHELS linkage has shown lower directly age-standardised rates and ratios of colorectal cancer in the South Asian population in Scotland (especially in Pakistani men), as well as in Chinese men,[19] linkage of census data with cancer registry has allowed us to examine test positivity, pathology and cancer outcomes by ethnic groups where available.

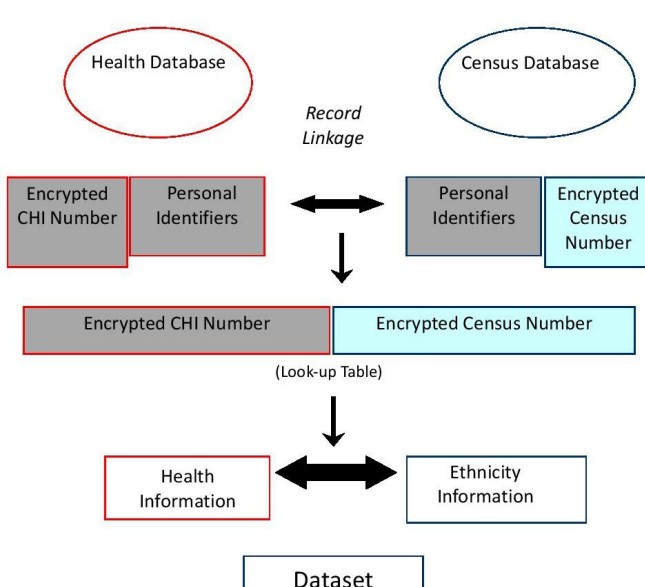

**Figure 1** Scottish health and ethnicity linkage study— linkage of health and census datasets. CHI, Community Health Index.

## METHODS
### Data linkage
Methods of SHELS retrospective cohort studies have been published.[16 17] We followed a strict protocol that preserved anonymity and maintained separation of personal data from the Census and NHS, and clinical data (figure 1). SHELS used computerised, probability matching of names, addresses, sex and dates of birth to link the Census 2001 for Scotland, to the CHI, which is a register of patients using the NHS. This created a file containing the linked encrypted CHI and encrypted census numbers for a cohort of 4.62 million people (95% of those completing the census and 90% of the estimated Scottish population in 2001). We used this file to link census variables to a previously linked Scottish Bowel Cancer Screening Programme (SBCSP) and Scottish Cancer Registry (SMR06) database.

### Ethnicity and religious data
The Scottish Census 2001 provided ethnic group as reported by either individuals or the householder completing the form based on a question followed by a choice of 14 categories. Unless stated otherwise, we have used the official categories, capitalising them as in census reports. Ethnic group is a legally required field that was well completed (95.8%) and, after imputation (4.3%), available for 100% of those completing the census form.[20] If necessary because of small numbers, we aggregated the Bangladeshi group with the Other South Asian group; and the Caribbean, African and Black Scottish or other Black groups into one African origin group in order to comply with data release stipulations of the data controller. Any mixed background is one of the distinct ethnic categories in the Census, designed for use by people who perceive themselves as belonging to more than one ethnic group,

usually with each parent in a different ethnic group. Following our analytical strategy, ethnic groups were only omitted to avoid potential disclosure of identity. We did not report results for the 'all other' ethnic group as this is an exceptionally diverse group of people and it is difficult to interpret results in any meaningful way. Religion was recorded on the Scottish Census 2001 in specific categories based on both self-reported current religion and self-reported religion of upbringing.

## Screening uptake

Individuals aged 50–74 years are invited to participate in bowel screening in Scotland every 2 years (a screening 'round'). Analyses were restricted by age to 50–74 years as the age range invited to participate in the screening programme, but we also examined screening uptake in the over 75s who chose to 'opt-in'.[2] Uptake of bowel screening was defined as a completed screening round using the FOBt (ie, screenee received a positive or negative test result).

## Sociodemographic data

Census data included age, sex, country of birth (UK/Ireland (RoI) born or born outside UK/RoI) and socioeconomic status (SES). Four socioeconomic indicators were used: (1) the postcode-based Scottish Index of Multiple Deprivation, (2) highest qualification of the individual, (3) a combined measure of highest qualification (individual level for people aged 16–74 and household level for children and elderly, as individual data are not collected for these groups) and (4) household tenure.

Ninety-nine per cent of the White Scottish group, 50% of the Indian group, 59% of the Pakistani group, 42% of the Other South Asian group, 41% of the African origin group, 36% of the Chinese group and 28% of the other white group were born in the UK/RoI in our linked census database.

## Outcomes

We primarily analysed uptake (persons successfully completing a kit and getting a final result that is, an outright positive or negative result) of bowel cancer screening between 2007 and 2013 in Scotland. First and second round (ie, where eligible participants are invited every 2 years) of screening were analysed separately. We further analysed the rate of positive screening test results in this participating population, and bowel cancer detection rates. The cohort of screening invitees analysed were those included in the Scottish Census 2001 who subsequently were still living in Scotland at the time of screening invitation. For analyses of screen detected invasive cancer, round 1 and round 2 (ie, where eligible participants are invited every 2 years) data were combined: round two figures include many of the same people as in round 1 results plus some newly entering the eligible age group, and who were resident at the 2001 Census.

## Data analysis

We followed a prespecified analysis plan (https://www.ed.ac.uk/usher/scottish-health-ethnicity-linkage/key-information). We calculated, for each outcome, by sex and ethnic group: uptake in screening in both round 1 and round 2; age-adjusted rates per 100 000 population; risk ratios (RRs) and their 95% CI using Poisson regression with robust variance adjusted for age and subsequently adjusted for SES and country of birth. We multiplied the estimates by 100 to facilitate the interpretation of the results as percentages, as per the SHELS policy and analysis plan. We adopted a previously published approach for choosing variables that were potential confounding showing consistency across ethnic groups.[16] Two SES indicators (household tenure and combined qualification) were consistently associated with the outcome across ethnic groups. The standard reference population was the White Scottish population. We also compared uptake rates by religion separately for men and women.

Data were analysed using SAS V.9.4 (SAS).

## Limited availability of Grampian data

For technical reasons, data on pathology (polyps, adenoma, cancer) and invasive cancer were unavailable from Grampian Health Board.

## Ethics and disclosure

Ethical and other permissions and related issues for SHELS methodology have been reported in detail including an independent assessment of SHELS' approach by an ethicist.[16 17 21] To comply with the Data Protection Act and safe-setting rules the data set only contained specific disease outcomes. Other outcomes were excluded to minimise risks of inadvertent disclosure of identity. The analysis was conducted on a standalone computer in a locked room in the National Records of Scotland (NRS), by named researchers with appropriate clearance and training (LW, GC and MS) and following a strict disclosure protocol.

Outputs leaving the safe setting as well as this manuscript were reviewed by the NRS Disclosure Committee. The analysis was done on exact numbers. However, the released numerators and denominators were rounded to the nearest 5.

Authors developed a Directed Acyclic Graph to aid the interpretation of results and help generate areas for further investigation (online supplemental figure 1).

## Patient and public involvement

SHELS established a Public Engagement Panel, comprised of a mix of ethnic groups, sexes and ages. This Public Engagement Panel provided patient and public involvement perspectives on SHELS methodological approach, including the research questions and design of this study. At the end of the study, results were shared with the Panel who commented on the findings and contributed to the dissemination plan.

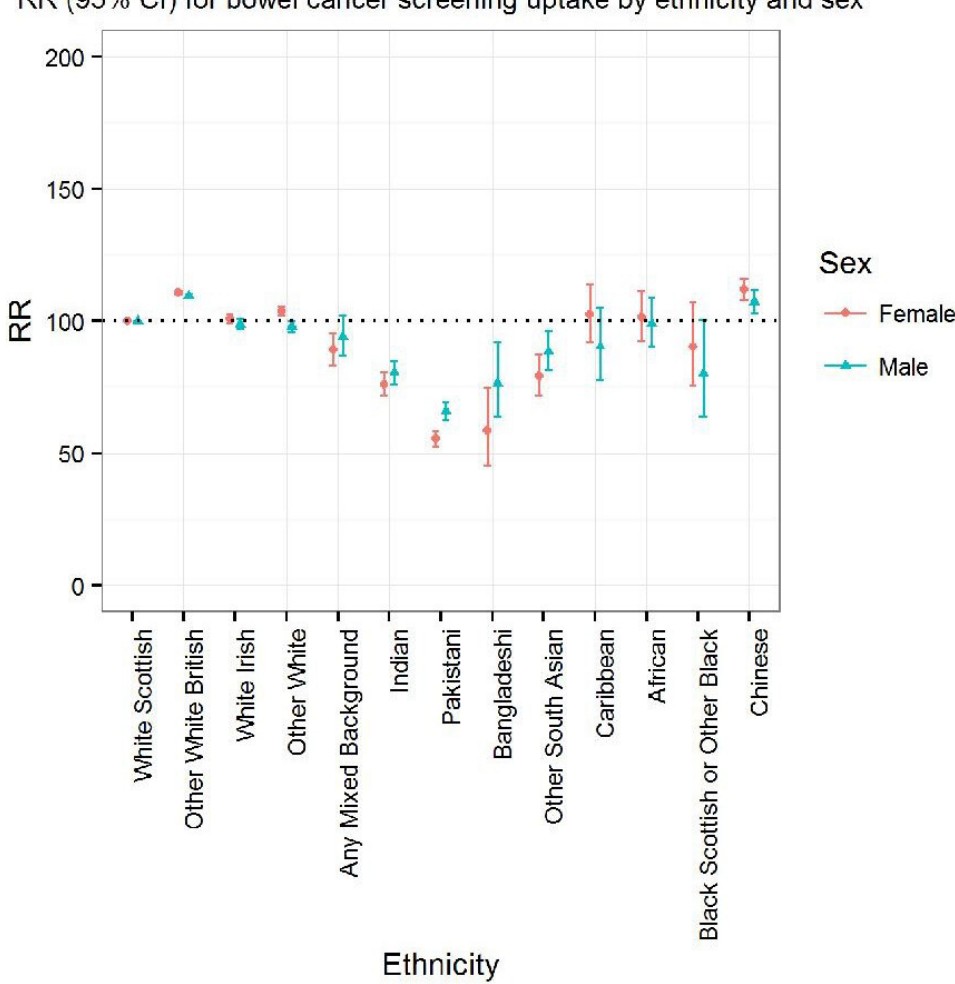

**Figure 2** Bowel cancer screening uptake by ethnicity for round 1 relative to the white Scottish population: age-adjusted risk ratios (RRs).

## RESULTS

### Linkage

Linked data were available for 1 666 575 of 1 926 060 individuals invited to participate in round 1 screening, a linkage rate of 86.5%. Of the 1.67 million matched at round 1, 1 407 835 individuals were invited to round 2. We present here round 1 results, with round 2 results and additional analyses available in online supplemental tables 1A, B, 2A, B, 3A, B, 4A, B, 5A, B, 6A–D,7A, B.

### Uptake of bowel screening by ethnic group

Uptake in specific ethnic groups were compared with the White Scottish population, unless specified otherwise. Figure 2 shows bowel cancer screening uptake in men and women for round 1 by ethnic group. For men, age-adjusted RRs were higher in the Other White British (RR (95% CI) 109.6 (108.8 to 110.3)) and Chinese (107.2 (102.8 to 111.8)) groups as they were more likely to return their kit once invited to screening compared with White Scottish men. Uptake was comparatively lower in other ethnic groups and especially so in Indian (80.5 (76.1 to 85.1)), Pakistani (65.9 (62.7 to 69.3)), Bangladeshi (76.6 (63.9 to 91.9)) and other South Asian (88.6 (81.8 to 96.1))

men. Further adjustment for UK/RoI-birth and SES did not greatly alter the associations apart from adjustment for UK/RoI-birth in Chinese men making their uptake converge towards the levels of uptake of White Scottish men (online supplemental table 1A).

Similarly in women (figure 2), age-adjusted RRs were higher in both Other White British (110.9 (95% CI 110.2 to 111.6)) and Chinese (112 (95% CI 108.2 to 115.9) women compared with White Scottish women, and again uptake was comparatively lower in women from Indian (76.1 (95% CI 72 to 80.5)), Pakistani (55.5 (95% CI 52.5 to 58.8)), Bangladeshi 58.5 (95% CI 45.6 to 75.1)) and other South Asian (79.3 (95% CI 72 to 107.2)) ethnic groups. Further adjustment for UK-birth and SES did not alter the associations observed (online supplemental table 1B).

Screening uptake rates by ethnic group for round 2 showed similar patterns for both men and women.

### Uptake of bowel screening by current religion and religion of upbringing

Figure 3 shows bowel cancer screening uptake in men and women for round 1 by self-reported current religion

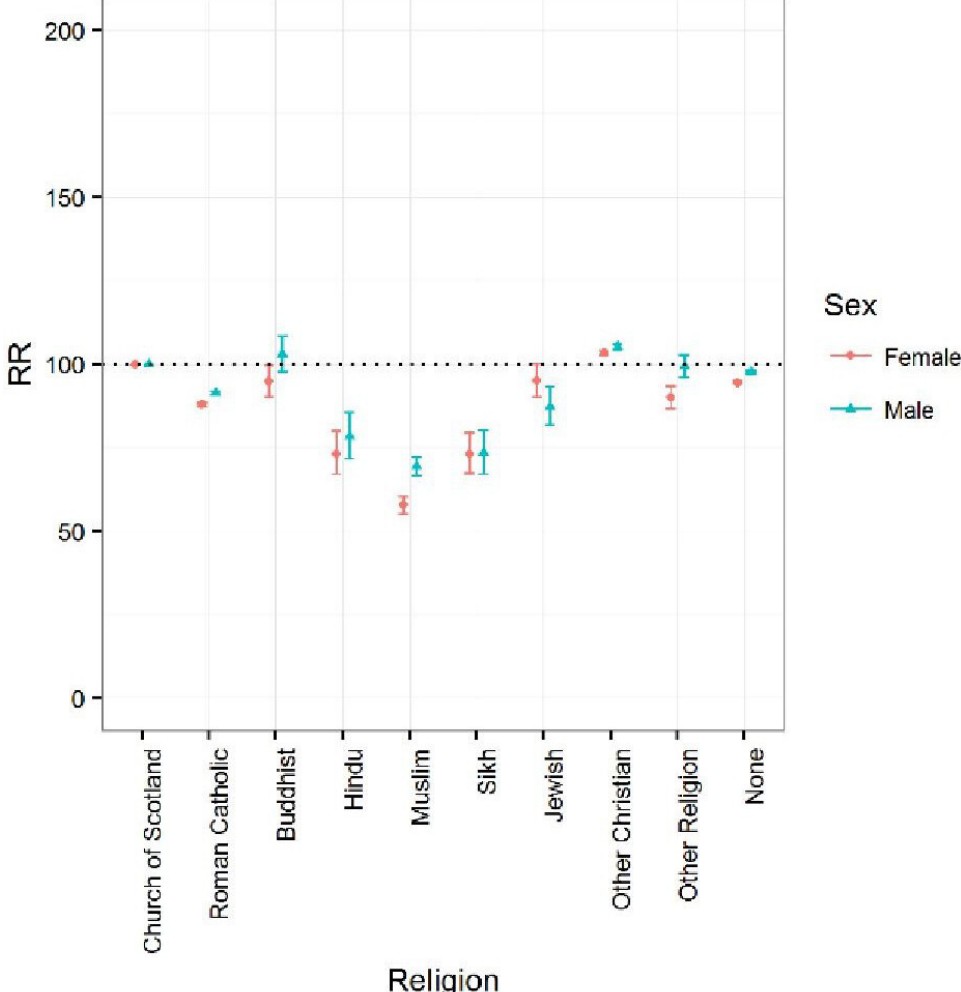

**Figure 3** Bowel cancer screening uptake by religion for round 1 relative to the reference population (Church of Scotland): age-adjusted risk ratios (RRs). Campbell *et al* Ethnic and religious variations in bowel cancer screening in Scotland.

as in the 2001 Census. Age-adjusted RRs were lower among Hindu (RR (95% CI) 78.4 (71.8 to 85.6)), Muslim (69.5 (95% CI 66.7 to 72.30), and Sikh (73.4 (95% CI 67.1 to 80.3)) men compared with those who identified current religion as Church of Scotland. Smaller differences compared with the reference population were observed among those who reported being Jewish (87.3 (95% CI 1.8 to 93.2)), Roman Catholic (91.4 (95% CI 90.8 to 92)), or none (no religion) (97.7 (95% CI 97.2 to 98.2)). Further adjustment for UK/RoI-birth and SES did not alter the trends observed.

In women, there was generally a lower uptake across groups compared with those who identified current religion as Church of Scotland apart from the Other Christian, with age-adjusted RRs being lower among Hindu (73.2 (95% CI 67 to 80), Muslim (57.8 (95% CI 55.2 to 60.5)), and Sikh (73.2 (95% CI 67.4 to 79.5)) women compared with the reference Church of Scotland population. Age-adjusted RRs for Roman Catholic women were lower (87.9 (95% CI 87.4 to 88.4)) compared with the reference Church of Scotland populations; further

adjustment for UK-birth and SES only modestly reduced the differences.

Screening uptake rates by religion of upbringing for men (online supplemental table 2A) and women (online supplemental table 2B) showed overall similar patterns.

Screening uptake in round 2 for both current religion and religion of upbringing showed similar patterns.

### Positivity of bowel screening test by ethnic group

Table 1 shows that age-adjusted RRs for positivity of FOBt were lower in Other White British (60.5 (95% CI 56.1 to 65.3)) men compared with White Scottish men in round 1. For women, positivity by ethnic group in round 1 (table 1) also showed lower positivity for the other White British group (67.3 (95% CI 61.6 to 73.5)) compared with White Scottish women. However, for selected other ethnic groups including Indian and Pakistani women there is some indication of lower test positivity rates compared with White Scottish women but CIs were wide due to the small sample size and straddling the reference value of 100. Similar patterns were seen in round 2 for both men and women.

**Table 1** (A) Positive screen test results (round 1) by ethnic group in men: age-adjusted rates and risk ratios (RRs). RRs are adjusted for age, UK/RoI-born (vs born outside UK/RoI) and socioeconomic status (household tenure and combined individual and household level education) with 95% CIs. RRs (95% CIs): adjustment.(B) Positive screen test results (round 1) by ethnic group in women: age-adjusted rates and RRs. RRs are adjusted for age, UK/RoI-born (vs born outside UK/RoI) and socioeconomic status (household tenure and combined individual and household-level education) with 95% CIs. RRs (95% CIs): adjustment

| Ethnic group | Positive screen test results | Completed screen kit returned | Rates/100 000 | Age | Age and UK/RoI-born | Age and two socioeconomic variables | Age, UK/RoI-born and two socioeconomic variables |
|---|---|---|---|---|---|---|---|
| **(A)** | | | | | | | |
| **Men** | | | | | | | |
| White Scottish | 11100 | 362865 | 3060 | 100 | 100 | 100 | 100 |
| Other White British | 685 | 37040 | 1844 | 60.5 (56.1 to 65.3) | 60.9 (56.4 to 65.7) | 68.9 (63.8 to 74.5) | 69.2 (64.1 to 74.7) |
| White Irish | 130 | 4220 | 3081 | 98.6 (83.2 to 116.8) | 98.7 (83.3 to 116.9) | 99.5 (84 to 117.8) | 99.5 (84 to 117.9) |
| Other White | 110 | 4045 | 2770 | 96 (80 to 115.2) | 111 (90.3 to 136.5) | 104.6 (87.2 to 125.5) | 115.3 (93.3 to 142.5) |
| Any mixed Background | 10 | 310 | 2913 | 101.6 (53.1 to 194.5) | 111.8 (58.4 to 213.9) | 108.3 (56.8 to 206.6) | 114.7 (60.2 to 218.7) |
| Indian | 15 | 705 | 1844 | 58.7 (34.3 to 100.5) | 71.6 (41.1 to 124.6) | 67.6 (39.5 to 115.7) | 76.9 (44.1 to 134) |
| Pakistani | 20 | 1015 | 2172 | 74.8 (49.9 to 112.3) | 92 (59.7 to 141.6) | 74.5 (49.6 to 111.7) | 85.3 (55.3 to 131.6) |
| Other South Asian | 10 | 400 | 2764 | 105.2 (59.1 to 187) | 126.9 (70.4 to 228.6) | 113.2 (64 to 200.2) | 127.3 (71.2 to 227.7) |
| African origin | 10 | 360 | 2500 | 97.4 (51.1 to 185.7) | 114.7 (59.4 to 221.2) | 102.5 (53.8 to 195.4) | 113.7 (58.9 to 219.4) |
| Chinese | 30 | 990 | 3128 | 114.2 (80.9 to 161.3) | 141.1 (96.7 to 205.7) | 109.4 (77.4 to 154.5) | 125.7 (86 to 183.7) |
| **(B)** | | | | | | | |
| **Women** | | | | | | | |
| White Scottish | 8015 | 444425 | 1803 | 100 | 100 | 100 | 100 |
| Other white British | 510 | 42950 | 1187 | 67.3 (61.6 to 73.5) | 67.3 (61.5 to 73.5) | 76.4 (69.9 to 83.5) | 76.2 (69.7 to 83.3) |
| White Irish | 80 | 5255 | 1542 | 82.8 (66.7 to 102.7) | 82.8 (66.7 to 102.7) | 86.2 (69.5 to 107) | 86.2 (69.5 to 107) |
| Other white | 90 | 5840 | 1523 | 86.9 (70.7 to 106.8) | 86.8 (67.1 to 112.3) | 98.8 (80.3 to 121.5) | 92.8 (71.2 to 120.8) |
| Any mixed Background | 10 | 435 | 2771 | 169.8 (96.7 to 297.9) | 169.6 (95.7 to 300.7) | 173.6 (99.4 to 303.4) | 167.5 (94.9 to 295.4) |
| Indian | 10 | 690 | 1744 | 100.8 (58.2 to 174.5) | 100.6 (57.1 to 177.3) | 110.5 (63.8 to 191.4) | 102.6 (58.1 to 181.3) |
| Pakistani | 15 | 870 | 1724 | 106.5 (64.6 to 175.5) | 106.2 (62.9 to 179.3) | 98.6 (59.8 to 162.6) | 91.2 (53.9 to 154.5) |
| Other South Asian | . | 265 | | | | | |
| African Origin | 10 | 395 | 1515 | 98.2 (44.3 to 217.6) | 98.1 (44 to 218.7) | 107.2 (48.5 to 236.9) | 101.1 (45.4 to 225.2) |
| Chinese | 25 | 1210 | 2066 | 130.6 (88.9 to 191.8) | 130.3 (85 to 199.9) | 125.2 (85.4 to 183.5) | 115.2 (75 to 177) |

**Table 2** Screen detected invasive cancer by ethnic group (rounds 1 and 2; men and women combined): age-adjusted rates and risk ratios (RRs)

| Ethnic group | Cancers | Invited into screening | Rates/ 100 000 | Age | Age and UK/RoI-born | Age and two socioeconomic variables | Age, UK/RoI-born and two socioeconomic variables |
|---|---|---|---|---|---|---|---|
| White Scottish | 2025 | 2 428 585 | 83.4 | 100 | 100 | 100 | 100 |
| Other White British | 145 | 205 420 | 70.5 | 84 (71 to 99.3) | 85 (71.8 to 100.5) | 79.5 (67.1 to 94.1) | 80.4 (67.9 to 95.2) |
| White Irish | 25 | 29 770 | 77.3 | 86.5 (57.4 to 130.4) | 86.6 (57.5 to 130.5) | 86.7 (57.5 to 130.6) | 86.7 (57.5 to 130.7) |
| Other white | 20 | 28 620 | 71.2 | 92.5 (59.6 to 143.7) | 130 (71.5 to 236.4) | 88.9 (57.2 to 138) | 127.1 (70.4 to 229.7) |
| * | | | | | | | |

Results exclude Grampian health board. RRs are adjusted for sex, age, UK/RoI-born (vs born outside UK/RoI) and socioeconomic status (household tenure and combined individual and household level education) with 95% CIs. RRs (95% CIs): adjustment.
*Results for any mixed background, South Asian (Indian, Pakistan or other) or Chinese ethnic groups are not provided as they are so few as to be potentially disclosive (see the Methods section).

## Bowel cancer detection and pathology by ethnic group

Table 2 shows bowel cancer detection rates via the screening test by ethnic group, for rounds 1 and 2, and for men and women combined, as this was necessary given the small numbers. Compared with the White Scottish population, other White British individuals had a lower age-adjusted RR of a diagnosis of screen-detected invasive cancer (84 (95% CI 71 to 99.3)); this result was not greatly altered after adjustment for UK-born and SES. Over the two rounds of screening, the number of invasive cancers found in individuals from other ethnic groups were too small to report for the risk of disclosure.

Online supplemental table 3A,B shows age-adjusted rates and RRs for pathology detected, for polyps, adenomas and cancer combined, for men and women, respectively. In comparison to the White Scottish population, numbers were small in each of the other ethnic groups. Only for Pakistani men was a lower rate of pathology detected (64.5 (95% CI 42.5 to 97.7)) compared with White Scottish men.

## Uptake of bowel screening in older individuals

Individuals aged 75 and older are able to opt in to bowel screening in Scotland. Table 3 shows age-adjusted RRs for screening uptake by ethnic group for men and women, respectively. Chinese men had higher uptake (112.8 (95% CI 113.3 to 114.3)) compared with White Scottish men (table 3), as did Chinese women compared with White Scottish women (table 3: 116.7 (95% CI 115 to 118.5)). Adjustment for SES did not greatly affect this association in either Chinese men or women, however, further adjustment for UK-birth in Chinese men the RR converged towards that of white Scottish men.

## DISCUSSION
## Summary of findings

Although ethnic variation in colorectal screening uptake is increasingly recognised internationally,[22] detailed description in relation to specific ethnic groups is lacking. We report complex patterns of variation in colorectal cancer screening uptake by ethnic group in Scotland,

with pronounced lower screening uptake among the South Asian groups compared with the White Scottish population, and higher uptake among the Chinese and other White British populations. We found little variation by ethnicity in later stages of the screening process.

## Strengths and limitations of the study

Our results are to our knowledge the most fine-grained analyses of bowel screening uptake by ethnicity reported to date, and with a nationally tested classification of ethnic groups. For the first time, national Scottish Census 2001 data were used to show differences in uptake for separate Indian, Pakistani and Bangladeshi groups, for separate White groups, and for the first time showing uptake among Caribbean, African and Chinese groups as well as by religious groups. Additionally, SHELS benefits from high overall linkage rates of census and NHS CHI numbers (95%), with a large national population (4.62 million people), and in this study a high linkage rate with the Bowel Screening data (86%). However, we acknowledge that the small numbers of outcomes for some non-white populations has required aggregation of heterogeneous ethnic groups; for example, African, Caribbean, black, black Scottish or black British. For invasive cancers, we were unable to report on some ethnic groups due to reporting restrictions on potentially disclosive numbers (table 3). Given the constraints of data release for reasons of patient confidentiality, understanding patterns of uptake in some ethnic groups will require additional research in other settings where numbers within distinct ethnic groups are sufficiently large.

We are reporting on 2001 Census data. Immigration to Scotland over the last 18 years has affected the distribution of ethnic groups within Scotland[23 24]: in particular, we do not report on bowel screening uptake among the Polish population, now one of Scotland's largest ethnic groups, where breast screening uptake is low.[25] Such analyses are not possible routinely and require a new study with linkage of bowel screening data to the 2011 Census. Nonetheless, the results reported here provide important

**Table 3** (A) Bowel cancer screening uptake (round 1) by ethnic group in men aged 75 years and over: age adjusted rates and risk ratios (RRs) RRs are adjusted for UK/RoI-born (vs born outside UK/RoI) and socioeconomic status (household tenure and combined individual and household level education) with 95%. RRs (95% CIs): adjustment.(B) Bowel cancer screening uptake (round 1) by ethnic group in women aged 75 years and over: age-adjusted rates and RRs. RRs are adjusted for UK/RoI-born (vs born outside UK/RoI) and socioeconomic status (household tenure and combined individual and household level education) with 95% CIs. RRs (95% CIs): adjustment

| Ethnic group | Completed screen kit returned | Requested screening | Rates/100 000 | Unadjusted | UK/RoI-born | Two socioeconomic variables | UK/RoI-born and two socioeconomic variables |
|---|---|---|---|---|---|---|---|
| **(A)** | | | | | | | |
| Men | | | | | | | |
| White Scottish | 2460 | 2780 | 88489 | 100 | 100 | 100 | 100 |
| Other White British | 180 | 215 | 84186 | 95 (89.5 to 100.8) | 94.7 (89.2 to 100.5) | 94.4 (88.8 to 100.3) | 94.2 (88.6 to 100) |
| White Irish | 30 | 30 | 87500 | 98.7 (86.6 to 112.6) | 98.5 (86.4 to 112.3) | 98.5 (86.3 to 112.4) | 98.3 (86.1 to 112.2) |
| Other White | 20 | 25 | 88000 | 99.3 (85.9 to 114.8) | 93.2 (78.9 to 110.1) | 98.2 (85 to 113.4) | 92.6 (78.6 to 109.2) |
| Any Mixed Background | | | | | | | |
| South Asian | 15 | 20 | 76190.5 | 86 (67.7 to 109.2) | 78.3 (60.8 to 100.8) | 85.1 (67.3 to 107.7) | 78.1 (60.8 to 100.4) |
| Chinese | 10 | 10 | 100000 | 112.8 (111.3 to 114.3) | 102.3 (93.6 to 111.7) | 115.1 (110.3 to 120.1) | 105.2 (95.3 to 116) |
| **(B)** | | | | | | | |
| Women | | | | | | | |
| White Scottish | 2470 | 2885 | 85615.3 | 100 | 100 | 100 | 100 |
| Other White British | 170 | 195 | 87113.4 | 101.7 (96.1 to 107.5) | 101.6 (96.1 to 107.5) | 100.9 (95.3 to 106.8) | 100.9 (95.3 to 106.8) |
| White Irish | 30 | 30 | 90322.6 | 105.4 (93.9 to 118.4) | 105.4 (93.9 to 118.4) | 105.7 (94.1 to 118.9) | 105.7 (94.1 to 118.9) |
| Other White | 40 | 40 | 95000 | 110.9 (103.1 to 119.2) | 110.3 (99.7 to 122.2) | 110.3 (102.5 to 118.7) | 109.9 (99.2 to 121.8) |
| Any Mixed Background | | | | | | | |
| South Asian | 10 | 15 | 64705.9 | 75.5 (53.1 to 107.3) | 75.1 (52.2 to 107.9) | 75.3 (53.1 to 106.8) | 75 (52.3 to 107.6) |
| Chinese | 10 | 10 | 100000 | 116.7 (115 to 118.5) | 116 (104.3 to 129) | 117.7 (114.5 to 121) | 117.2 (105.1 to 130.7) |

insights into recent uptake patterns and set a benchmark for any future variation in bowel screening uptake rates as the population profile changes.

Finally, we recognise that the reported screening uptake rates are descriptive and not explanatory. We adjusted for determinants of ethnic inequalities in bowel screening such as SES and UK birth but these made little difference to the patterns observed. Further potential mechanisms need to be explored, including cultural and religious beliefs, and the influence (if any) of knowledge of or exposure to screening programmes in other health systems (see online supplemental figure 1 for potential variables influencing participation).

Data on a number of variables (pathology (polyps, adenoma, cancer) and invasive cancers) were unavailable from Grampian Health Board: sensitivity analyses (available on request) indicates that approximately 10%–12% of the denominator in the Scottish population were missing for these variables in the Scottish population. Grampian Health Board comprises only 10.1% of the Scottish population, and with a non-white Scottish population of 15% compared with 12% in Scotland overall, there is no reason to expect that inclusion of these data would have altered the observed patterns in (table 2) .[26] Data on uptake rates were complete.

### Existing literature

We found lower rates of screening uptake in South Asian populations, reflected in both ethnic group and current religion. Lower screening uptake among South Asian communities in the UK has been a feature of the screening programme since its inception[8 9 12] and the factors influencing this are increasingly being understood. Many factors such as lack of awareness and understanding of the purpose of screening, and fear and fatalism about cancer are seen across all ethnic groups.[6 27–30] However, limitations with English-language screening materials (translated materials often require request), the need to rely on younger family members, cultural difficulties associated with handling of faeces and social norms are additional barriers among South Asian ethnic groups.[7 30] Differences in breast screening uptake by ethnic group in Scotland have been reported previously by our group (higher non-attendance rates to breast cancer screening among Pakistani, black, other South Asian and Indian women),[18] as has variation in relation to numerous other health outcomes.[31–33]

As mentioned, lower directly age-standardised rates and ratios of colorectal cancer in the South Asian population have been reported in Scotland.[19] The RRs we report here suggesting lower RRs of FOBt positivity in Indian and Pakistani men (table 1) are consistent with this, although need to be interpreted with care due to wide CIs. Lower colorectal cancer rates in some ethnic communities may result in less perceived personal relevance and hence tailored educational interventions will need to acknowledge lower colorectal cancer rates while also addressing the identified barriers and facilitators.[34–37]

There is a need for open discussion within bowel screening programmes and policy making of potentially variable benefits for different ethnic groups of screening uptake. The lower uptake rates may be appropriate for some groups, and genuine informed consent may require acknowledgement that some have less to gain in terms of absolute risk reduction. At a programmatic level, there is a balance between lower cancer risk and uptake of screening, and further work is warranted to address how issues of equality of access, cost-effectiveness and effectiveness are maintained. Although at a population level, the risk may be lower, messages aimed at the individual level need to communicate clearly the potential advantages of screening uptake within an informed choice framework.

The relatively high uptake rates among both Chinese men and women compared with the White Scottish men and women were unexpected, and not previously recognised in the Scottish population. Bansal et al found that age-adjusted RRs for breast screening uptake were similar among Chinese women compared with White Scottish women.[18] High FOBt positivity rates were observed in both Chinese men and women in both rounds 1 and 2; this is despite the lack of evidence of higher incidence of colorectal cancer in the Chinese community in earlier SHELS work.[19] Further research is warranted, not only to determine if these findings can be replicated in other Chinese communities in the UK, but also to explore any cultural or other factors underlying high screening uptake. Low awareness of colorectal cancer screening was found among Chinese participants in an EthniBus survey.[4] Importantly, though, as noted above low rates of colorectal screening uptake (by flexible sigmoidoscopy) have been reported in areas of high non-white ethnicity but these were not broken down by ethnic group.[38]

While numbers were relatively small, only limited variation in colorectal screening uptake was seen in the over 75 population; there is, however, some indication that South Asian men and women were less likely to opt-in. This is a self-selecting group of individuals who are likely to differ from their peers in terms of other health behaviours, motivation and levels of comorbidity. The low overall number of opt-ins is consistent with findings from the Bowel Screening Pilot in England.[8]

### Implications for policy and practice

Addressing observed inequalities in screening uptake will require multifaceted interventions. Telephone-based interventions have been shown to increase colorectal screening uptake in ethnically diverse areas of London[39] but have resource implications. Patient navigators have been shown to be effective in some settings.[40] Further exploratory work and engagement with local communities is needed to develop, refine and test culturally appropriate interventions with salience to different ethnic groups; critically, these must ensure principles of informed choice are respected and incorporated throughout.[41] Our reported variations in uptake by religion are, seemingly, novel: in particular, the lower uptake

among Roman Catholic populations compared with the reference population, persisting even when adjusted for socioeconomic variables, is puzzling. Addressing such variation by religion may be amenable to targeted faith-based interventions.[42] Others have found variable influence of religiosity on screening uptake, with social support only partially mediating the relationship between religiosity and bowel screening uptake.[43] Comparing facilitators and barriers across groups may provide fresh insight into potential interventions.[44] Further, the introduction of the FIT in the SBCSP in late 2017 has been shown to increase overall screening uptake,[2] this provides an impetus to monitor the impact within ethnic groups over time (work currently underway by authors).

**Author affiliations**
[1]Usher Institute, The University of Edinburgh, Edinburgh, UK
[2]Population and Health research group, School of Geography and Sustainable Development, University of St Andrews, St Andrews, UK
[3]Public Health Scotland, Edinburgh, UK
[4]Institute of Health & Wellbeing, University of Glasgow, Glasgow, UK
[5]Public Health Scotland, Glasgow, UK
[6]Surgery and Molecular Oncology, University of Dundee, Dundee, UK
[7]School of Medicine, Department of Child Health, University of Aberdeen, Aberdeen, UK

**Contributors** All authors meet authorship criteria. RB was the PI of SHELS, and CC was the PI of this specific component of the research. RB, CC, AD, DHB and DW conceived the study and planned it along with KR, GS and RS. LW, GC and MS carried out data analysis. DB provided statistical advice. All authors contributed to the writing of the paper. CC is the guarantor.

**Funding** This work was supported by the Chief Scientist's Office (grant number CZH/4/878), Cancer Research UK (grant number C3743/A16594), and supplementary funding from NHS Health Scotland. ISD and National Records of Scotland both made 'in-house' contributions to the work.

**Competing interests** None declared.

**Patient consent for publication** Not required.

**Ethics approval** The work was approved by the Multicentre Research Ethics Committee for Scotland (16/12/2013, Scotland A REC Ref: 13/SS/0225) and the Privacy Advisory Committee of NHS National Services Scotland. Caldicott Guardian approval was obtained for access to SBCSP data. The study was performed in accordance with the Declaration of Helsinki.

**Provenance and peer review** Not commissioned; externally peer reviewed.

**Data availability statement** Data may be obtained from a third party and are not publicly available. Researchers who wish to access SHELS data should apply to National Records of Scotland (https://www.nrscotland.gov.uk/) and ISD (http://www.isdscotland.org/). They are maintained in a secure environment and governed by ethical and other restrictions on access.

**ORCID iDs**
Christine Campbell http://orcid.org/0000-0003-4868-0554
David H Brewster http://orcid.org/0000-0002-5346-5608

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
