## [Reviewer comments · BMJ Open]

ARTICLE DETAILS

TITLE (PROVISIONAL)	Are there ethnic and religious variations in uptake of bowel cancer screening? A retrospective cohort study among 1.7 million people in Scotland
AUTHORS	Campbell, Christine; Douglas, Anne; Williams, Linda; Cezard, Genevieve; Brewster, David; Buchanan, Duncan; Robb, Kathryn; Stanners, Greig; Weller, David; Steele, Robert; Steiner, Markus; Bhopal, Raj

VERSION 1 – REVIEW

REVIEWER	Henriette C. Jodal University of Oslo, Norway
REVIEW RETURNED	24-Mar-2020

GENERAL COMMENTS	Dear Editors, Thank you for the opportunity to review the manuscript «Ethnic and religious variations in uptake of bowel cancer screening among 1.7 million people in Scotland». The manuscript describes the variation in uptake among different ethnicities as well as religions in the first round of the FOBT colorectal cancer screening programme in Scotland. In addition, the authors present results for positive screen test results in round 1 and invasive cancers in round 1 and 2 combined, where they mainly show no differences between the ethnicities. My main concern is that the focus of the manuscript seems to be how to increase uptake in the groups where uptake is lower. Although it is important to give all ethnicities the same access to be screening, e.g. through information in different languages, the uptake in itself should not be the measure of a “successful” screening programme. The authors mention this shortly in the “Existing literature” section of the discussion, but the focus in the “Implications” section is only on how to increase uptake. The differences in uptake might be due to informed choice, influenced by religion and/or ethnicity, rather than lacking access to screening. Further, the authors describe the results in extension, but it is hard to grasp the main message as it disappears between the results from the secondary analyses. The secondary analyses mainly describe, as previously mentioned, no differences between ethnicities. I wonder why the authors choose to show so many tables and write so much about these outcomes in your methods, while they are not even mentioned in the “Summary of findings”. As the authors mention themselves, the results are only descriptive. I would like some more discussion about why the
---

	differences they find might be, and how they compare to different types of screening. The authors mention their previous studies on other types of cancer screening, but they do not mention or compare these results to their previous results. Are the differences they find a general trend in these ethnic groups? Have the authors looked at ethnicity and religion combined? Other comments:  - I always find it useful with a figure showing how the data was obtained, i.e. the linking of the data. - It is confusing that the authors use relative risk x 100 – in particular because they still call it the RR rather than 100xRR or similar. I think all readers of BMJ Open are used to RR in terms of 1, and this does make the manuscript less intuitive. - The authors use all the terms risk ratio, relative risk and rate ratio throughout the manuscript. Please choose one. - Table legends are long, and what this table shows disappears in the text. Consider writing what it shows at the beginning (e.g. Table 3a: “Bowel cancer screening uptake (round 1). Age-adjusted rates...”) - I think their choice to look at data from only Round 1 for some outcomes, while they use both Round 1 and 2 for some other outcomes, should be explained.
--	---

REVIEWER	Kaisa Fritzell NVS, Division of Nursing, Karolinska Universitetssjukhuset, Sweden
REVIEW RETURNED	05-May-2020

GENERAL COMMENTS	Thank you for the opportunity to review the manuscript entitled ‘Ethnic and religious variations in uptake of bowel cancer screening among 1.7 million people in Scotland’. This is an interesting article dealing with an important subject. I have, however, one major objection about the paper and that is the busy result section and the fact that no aim (only outcomes) is presented. For detailed comments, see below. BACKGROUND The background gives the rational for the study for the main outcomes but not for investigation colonoscopy findings, such as cancer. I would also have liked the background to problematize culture vs. religion Page 6, line 5 – please explain what you mean with the part ‘...self-selecting responders to...’ Page 7 – the aim is not clear to me since there is no aim or objective included in the background, please add. As far as I understand, the uniqueness with this study is, in comparison to other studies assessing uptake in ethnic groups, that you include ethnicity based on self-reported data. If I understood that right, it needs to be further clarified. Page 7, line 16-37 – this section is a bit confusing, it seems to be a mix of method, background and discussion. Please remove or place the parts were they belong. METHODS Ethnicity and religious data Page 8, line 44 – at first I thought that ‘Black Scottish’ was similar to being black and born in in Scotland. But, since you grouped them into ‘African origin’ I’m not sure. It seems odd to group
--

	individual on skin color instead of cultural background. Please clarify what you mean with 'Black Scottish'. Page 8, line 54 – clarify if religion was self-reported Socio-demographic data Page 8, line 27, (3) - it is not clear to me if 'measure of highest qualification' means the highest qualification in the family or household. Outcomes Page 9, line 52 – I wonder why you chose to include positive screening test and cancer as outcomes. The result section does not provide any impact on screening uptake, which seems to be the aim of the study. In addition, the background does not give the rational for that. Please clarify. This is important especially since the data on cancer seems limited. Data analyses Page 10, line 27 – remove 'our' in the beginning of the line Limited availability of Grampian data Page 10, line 44 – it is not clear to me what you mean with 'limited availability of Grampian data' please add what impact this limitation has on the study. This is especially important since the data is outcomes of the study. RESULTS The result section is busy reading with lot of tables, figures and supplements and a bit confusing with different grouping of ethnicity in different analyses. The result section might be easier to follow if aim and research questions were added and if all data from the supplements were presented more brief in the text. I also found it strange that you did analyses with groups of many different ethnic groups since that were one of your main objections (background) to previous studies. Tables and Figures. I can't find an explanation for 'Any Mixed Background' in the method section. DISCUSSION Summary of findings Page 16 – the content seem to be more of a conclusion to me. Page 16, limitations – I lack a discussion about the fact that you had to put different ethnicity groups together in several analyses and what impact that had on your results. Page 18, line 22 – this line may give the reader the rational for including colonoscopy findings, such as cancer. If that is the case, this should be introduced earlier in the paper, preferably in the background.
--	--

VERSION 1 – AUTHOR RESPONSE

Reviewer: 1

My main concern is that the focus of the manuscript seems to be how to increase uptake in the groups where uptake is lower. Although it is important to give all ethnicities the same access to be

screening, e.g. through information in different languages, the uptake in itself should not be the measure of a “successful” screening programme. The authors mention this shortly in the “Existing literature” section of the discussion, but the focus in the “Implications” section is only on how to increase uptake. The differences in uptake might be due to informed choice, influenced by religion and/or ethnicity, rather than lacking access to screening.

Thanks for raising this important point. We very much agree that the sole measure of a ‘successful’ screening programme should not be uptake alone: we too are committed to the principles of informed choice / decision-making in screening participation. However, at present there is evidence from the literature (as described in the manuscript) of lower awareness of cancer screening programmes among women in ethnic minority groups in the UK. This study sought to describe variation in bowel screening uptake in Scotland by ethnic minority group, and in the absence of incorporation of a validated informed choice measure of screening participation in the screening programme, we are not able to ascribe fully reasons for this. There may indeed for some screening invitees be a degree of informed choice influenced by religion and /or ethnicity, but equally there may be an un-informed choice due to lack of provision of culturally-informed materials that allow screening invitees to consider for themselves whether or not to accept an offer of screening.

We have amended wording in Implications to read: *‘Further exploratory work and engagement with local communities is needed to develop, refine, and test culturally-appropriate interventions with salience to different ethnic groups; critically, these must ensure principles of informed choice are respected and incorporated throughout.’⁴¹*

Further, the authors describe the results in extension, but it is hard to grasp the main message as it disappears between the results from the secondary analyses. The secondary analyses mainly describe, as previously mentioned, no differences between ethnicities. I wonder why the authors choose to show so many tables and write so much about these outcomes in your methods, while they are not even mentioned in the “Summary of findings”.

We thank both reviewers for the helpful and constructive comments regarding the number of results presented, including in the Supplementary Materials. We have streamlined the Results to ensure the main messages of the paper are clearer, focusing on presenting results from Round 1 in the manuscript. We now provide Round 2 data in Supplementary Materials; for full and transparent reporting against the data analysis plan we also provide results of additional analyses (Country of Birth, and Completion of Colonoscopy) in Supplementary Materials. We have added a sentence to the first paragraph of Results *‘We present here Round 1 results, with Round 2 results and additional analyses available in Supplementary Materials’*, and have modified wording referring to these in the remainder of the Results section. In the Summary of findings we have added *‘We found little variation by ethnicity in later stages of the screening process.’*

As the authors mention themselves, the results are only descriptive. I would like some more discussion about why the differences they find might be, and how they compare to different types of screening. The authors mention their previous studies on other types of cancer screening, but they do not mention or compare these results to their previous results. Are the differences they find a general trend in these ethnic groups?

Thanks for these helpful observations. Describing variation is an essential first step. Space constraints meant it was not possible to elaborate on the reasons for variation in screening in

the Discussion - we did provide a Directed Acyclic Graph (DAG) to aid the interpretation of results and help generate areas for further investigation (Supplementary Materials Figure 1). This was mentioned in Methods, but we then failed to refer to it directly in the Discussion. We have now added the following sentence to the Discussion (page 16) '*see Supplementary Figure 1 for potential variables influencing participation*'.

We have compared our results for bowel screening uptake with previous SHELS studies on breast screening (on page 17, in reference to uptake among the Chinese population). We have now provided additional wording: '*Differences in breast screening uptake by ethnic group in Scotland have been reported previously by our group (higher non-attendance rates to breast cancer screening among Pakistani, Black, Other South Asian and Indian women),¹⁸ as has variation in relation to numerous other health outcomes.³¹⁻³³*' (page 18)

Have the authors looked at ethnicity and religion combined?

No, we did not carry out these analyses – due to the overlap of ethnicity and religion, it would be difficult to interpret the findings. We recognise further qualitative work in this area is required, but is beyond the scope of this paper.

Other comments:

- I always find it useful with a figure showing how the data was obtained, i.e. the linking of the data.

Thanks for this helpful suggestion. We have now added a new Figure 1 (Scottish Health and Ethnicity Linkage Study – linkage of Health and Census datasets) and amended the remaining Figure numbers in the text and legends

- It is confusing that the authors use relative risk x 100 – in particular because they still call it the RR rather than 100xRR or similar. I think all readers of BMJ Open are used to RR in terms of 1, and this does make the manuscript less intuitive.

Thanks for this comment. The multiplication of the relative risks is part of our SHELS pre-specified analysis plan, recorded online (<https://www.ed.ac.uk/usher/scottish-health-ethnicity-linkage/key-information>). Within SHELS we are particularly keen to reach out to non-epidemiological audiences, particularly policymakers. They and the public understand percentages readily, and while within the epidemiological world decimal points are not a problem, they often are to the wider readership. All but one of approximately twenty SHELS papers have reported in exactly this way, including a number in the BMJ Open (e.g. Bhopal RS, Cezard G, Bansal N, Ward HJT, Bhala N; SHELS researchers. Ethnic variations in five lower gastrointestinal diseases: Scottish health and ethnicity linkage study. *BMJ Open*. 2014;4(10):e006120; Bansal N, Fischbacher CM, Bhopal RS, et al. Myocardial infarction incidence and survival by ethnic group: Scottish Health and Ethnicity Linkage retrospective cohort study. *BMJ Open*. 2013;3(9):e003415. Published 2013 Sep 13. doi:10.1136/bmjopen-2013-003415; Bhopal RS, Bansal N, Steiner M, Brewster DH; Scottish Health and Ethnicity Linkage Study. Does the 'Scottish effect' apply to all ethnic groups? All-cancer, lung, colorectal, breast and prostate cancer in the Scottish Health and Ethnicity Linkage Cohort Study. *BMJ Open*. 2012;2(5):e001957. Published 2012 Sep 25. doi:10.1136/bmjopen-2012-001957).

We trust our approach of reporting relative risks x 100, following both study protocol and BMJ Open precedent, remains acceptable to the Editors.

- The authors use all the terms risk ratio, relative risk and rate ratio throughout the manuscript. Please choose one.

Thanks for this comment. We have considered carefully our use of these terms throughout the manuscript, and now use the term risk ratio throughout

- Table legends are long, and what this table shows disappears in the text. Consider writing what it shows at the beginning (e.g. Table 3a: “Bowel cancer screening uptake (round 1). Age-adjusted rates...”)

Thanks. We have revised Table legends and Figure legends as suggested.

- I think their choice to look at data from only Round 1 for some outcomes, while they use both Round 1 and 2 for some other outcomes, should be explained.

As described above, we have streamlined results to focus on our main outcomes of interest, and have removed reporting of Round 2 data where this does not add to the findings. Where both Round 1 and Round 2 data are reported (now only in Table 2) this is to ensure sufficient data are available for analyses. We have added an explanatory sentence in Methods: *‘For analyses of screen detected invasive cancer, Round 1 and Round 2 (i.e. where eligible participants are invited every two years) data were combined’*

Reviewer: 2

Reviewer Name: Kaisa Fritzell

Institution and Country: NVS, Division of Nursing, Karolinska Universitetssjukhuset, Sweden Please state any competing interests or state ‘None declared’: None declared

Please leave your comments for the authors below Thank you for the opportunity to review the manuscript entitled ‘Ethnic and religious variations in uptake of bowel cancer screening among 1.7 million people in Scotland’. This is an interesting article dealing with an important subject. I have, however, one major objection about the paper and that is the busy result section and the fact that no aim (only outcomes) is presented. For detailed comments, see below.

BACKGROUND

The background gives the rationale for the study for the main outcomes but not for investigation colonoscopy findings, such as cancer. I would also have liked the background to problematize culture vs. religion Page 6, line 5 – please explain what you mean with the part ‘...self-selecting responders to...’

We have amended the last paragraph of the Background to read:

The primary aim of this paper was to describe bowel cancer screening uptake rates in detail by self-reported ethnic group, including White Scottish, Other White British, White Irish, Other White, Indian, Pakistani, Bangladeshi, Other South Asian, Caribbean, African, Other Black, Chinese, in addition to self-reported religious affiliation. Further, as previous SHELS linkage has shown lower directly age-standardised rates and ratios of colorectal cancer in the South Asian population in Scotland (especially in Pakistani men), as well as in Chinese men¹⁹, linkage of census data with cancer registry has allowed us to examine test positivity, pathology and cancer outcomes by ethnic group where available.

Culture vs religion – this is a complex topic, beyond the immediate scope of this paper. We have added words to the Background (page 6): *‘(while recognizing that ethnicity, religion and cultural background are over-lapping although not synonymous identities)’*.

Self-selecting responders – we used the term ‘self-selecting’ to indicate that not all responders to the survey described in reference 5 chose to complete the ethnicity question but acknowledge it is not clear and have removed the term ‘self-selecting’

Page 7 – the aim is not clear to me since there is no aim or objective included in the background, please add. As far as I understand, the uniqueness with this study is, in comparison to other studies assessing uptake in ethnic groups, that you include ethnicity based on self-reported data. If I understood that right, it needs to be further clarified.

The unique aspects of this work include linkage of the Census data to bowel cancer screening and cancer registry data sets, together with self-reported ethnicity data by detailed ethnic group. We trust the amended last paragraph of the Background makes this clearer.

Page 7, line 16-37 – this section is a bit confusing, it seems to be a mix of method, background and discussion. Please remove or place the parts where they belong.

We read this comment with interest. The last paragraph of the Background provides specific context and rationale for the study: a unique health and census data-linkage resource (the SHELS platform), previous findings using SHELS demonstrating variation in breast screening uptake by ethnic group, leading to us now using the same platform to examine variation in colorectal screening uptake by ethnic group. As described above we have amended wording to make the aims clearer, but otherwise we are satisfied these lines express the purpose of the study and lead the reader into the remainder of the paper.

METHODS

Ethnicity and religious data

Page 8, line 44 – at first I thought that ‘Black Scottish’ was similar to being black and born in in Scotland. But, since you grouped them into ‘African origin’ I’m not sure. It seems odd to group individual on skin color instead of cultural background. Please clarify what you mean with ‘Black Scottish’.

The 2001 Census question on Ethnic group classified people *‘according to their own perceived ethnic group and cultural background’*. Options in the 2001 Census included ‘Black, Black Scottish or Black British’, ‘Caribbean’ and ‘African’. The decision to group these together under ‘African origin’ due to potentially disclosive numbers was made in order to be consistent with Bansal et al 2012 (SHELS study on breast cancer screening and ethnicity), based on recent and distant ancestral origins in (sub-Saharan) Africa. We recognise the group ‘African origin’ is a simplification of complex cultural and experiential backgrounds, as well as multiple factors influencing self-identity. We have amended a sentence to the Discussion further acknowledging this limitation *‘However, we acknowledge that the small numbers of outcomes for some non-White populations has required aggregation of heterogeneous ethnic groups; e.g. African, Caribbean, Black, Black Scottish or Black British.’* We have also added a sentence *‘Given the constraints of data release for reasons of patient confidentiality, understanding patterns of uptake in some ethnic groups will require additional research in other settings.’*

Page 8 (actually page 7), line 54 – clarify if religion was self-reported

This sentence has been amended to:

'Religion was recorded on the Scottish Census 2001 in specific categories based on both self-reported current religion and self-reported religion of upbringing'.

Socio-demographic data

Page 8, line 27, (3) - it is not clear to me if 'measure of highest qualification' means the highest qualification in the family or household.

The 'combined measure of highest qualification' here applies at the individual level (in the Census it is not applicable to 'schoolchildren and full-time students living away from home during term time and all those under the age of 16')

Outcomes

Page 9, line 52 – I wonder why you chose to include positive screening test and cancer as outcomes. The result section does not provide any impact on screening uptake, which seems to be the aim of the study. In addition, the background does not give the rationale for that. Please clarify. This is important especially since the data on cancer seems limited.

Thanks for highlighting that the rationale for inclusion of positive screening test and cancer outcomes was not provided in the original submission. As described above, we have now amended the last paragraph of the Background to include reference to a previous SHELS study reporting lower age-standardised cancer rates among ethnic groups in Scotland.

Data analyses

Page 10, line 27 – remove 'our' in the beginning of the line

Thanks – we have corrected this typo

Limited availability of Grampian data

Page 10, line 44 – it is not clear to me what you mean with 'limited availability of Grampian data' please add what impact this limitation has on the study. This is especially important since the data is outcomes of the study.

Data on screening uptake, the primary outcome reported in this paper, were available from all health boards in Scotland, *including Grampian*. However, as mentioned, data on pathology (polyps, adenoma, cancer) and invasive cancer were unavailable for Grampian Health Board. This was due to a data processing issue that was subsequently rectified, but not in time for analysis: the screening data and clinical data were not matching correctly and thus the extract only had partial pathology available for Grampian patients.

We discuss this Limitation in the Discussion (page 16), with an additional sentence: 'Grampian Health Board comprises only 10.1% of the Scottish population, and with a non-White Scottish population of 15% compared to 12% in Scotland overall, there is no reason to expect that inclusion of these data would have altered the observed patterns in Table 2²⁶'

RESULTS

The result section is busy reading with lot of tables, figures and supplements and a bit confusing with different grouping of ethnicity in different analyses. The result section might be easier to follow if aim and research questions were added and if all data from the supplements were presented more brief in the text.

Thanks for this observation, also made by Reviewer 1. As described above, we have described the aims of the paper more clearly in the Background, and now provide Round 2 data in the Supplementary Materials with only brief mention in the Results section.

I also found it strange that you did analyses with groups of many different ethnic groups since that were one of your main objections (background) to previous studies.

As the reviewer has noted, for some analyses we do report data from combined ethnic groups. Combining ethnic groups in this way allowed us to have some precision in the estimates - otherwise the confidence intervals would be very wide indeed. In some cases combining data was a legal requirement where numbers in any one group are small (less than 5 individuals) in order not to lead to potential identification of any one individual. We have added a line to the Methods '*in order to comply with data release stipulations of the data controller*'; to make this clearer.

Tables and Figures. I can't find an explanation for 'Any Mixed Background' in the method section.

We have added (Methods, page 7): '*Any Mixed Background is one of the distinct ethnic categories in the Census, designed for use by people who perceive themselves as belonging to more than one ethnic group, usually with each parent in a different ethnic group*'

DISCUSSION

Summary of findings

Page 16 – the content seem to be more of a conclusion to me.

We have moved the sentence 'Differences in breast screening uptake by ethnic group in Scotland have been reported previously by our group,¹⁸ as has variation in relation to numerous other health outcomes.²²⁻²⁵' to elsewhere in the Discussion

Page 16, limitations – I lack a discussion about the fact that you had to put different ethnicity groups together in several analyses and what impact that had on your results.

We did acknowledge this as a specific limitation on page 16. However, we have now expanded this to read '*However, we acknowledge that the small numbers of outcomes for some non-white populations has required aggregation of heterogeneous ethnic groups; e.g. African, Caribbean, Black, Black Scottish or Black British*'. We have also added wording '*Given the constraints of data release for reasons of patient confidentiality, understanding patterns of uptake in some ethnic groups will require additional research in other settings where numbers within distinct ethnic groups are sufficiently large.*'

Page 18, line 22 – this line may give the reader the rationale for including colonoscopy findings, such as cancer. If that is the case, this should be introduced earlier in the paper, preferably in the background.

Thank you. As noted above, we have provided an additional sentence in the Background to prove rationale for test positive and cancer outcomes. We have retained Colonoscopy tables in the Supplementary Materials.

VERSION 2 – REVIEW

REVIEWER	Kaisa Fritzell NVS, Division of Nursing, Karolinska Institutet, Stockholm, Sweden
REVIEW RETURNED	09-Jul-2020
GENERAL COMMENTS	Dear authors, Thank you for taking all my comments seriously. This paper has really improved, and as far as I can see, it is ready for publication. Kindly Kaisa